# Predicting Rail Corrugation Based on Convolutional Neural Networks Using Vehicle’s Acceleration Measurements

**DOI:** 10.3390/s24144627

**Published:** 2024-07-17

**Authors:** Masoud Haghbin, Juan Chiachío, Sergio Muñoz, Jose Luis Escalona Franco, Antonio J. Guillén, Adolfo Crespo Marquez, Sergio Cantero-Chinchilla

**Affiliations:** 1Department of Structural Mechanics and Hydraulic Engineering, Andalusian Research Institute in Data Science and Computational Intelligence (DaSCI), University of Granada (UGR), 18001 Granada, Spain; jchiachio@ugr.es; 2Department of Materials and Transportation Engineering, Escuela Técnica Superior de Ingeniería, University of Seville, 41092 Seville, Spain; sergiomunoz@us.es (S.M.); escalona@us.es (J.L.E.F.); 3Department of Management, Complutense University of Madrid, 28040 Madrid, Spain; ajguillen@ucm.es; 4Department of Industrial Management, Escuela Técnica Superior de Ingeniería, University of Seville, 41092 Seville, Spain; adolfo@us.es; 5School of Electrical, Electronic and Mechanical Engineering, University of Bristol, Bristol BS8 1TR, UK; sergio.canterochinchilla@bristol.ac.uk

**Keywords:** rail corrugation, deep learning, convolutional neural networks, Grad-CAM

## Abstract

This paper presents a deep learning approach for predicting rail corrugation based on on-board rolling-stock vertical acceleration and forward velocity measurements using One-Dimensional Convolutional Neural Networks (CNN-1D). The model’s performance is examined in a 1:10 scale railway system at two different forward velocities. During both the training and test stages, the CNN-1D produced results with mean absolute percentage errors of less than 5% for both forward velocities, confirming its ability to reproduce the corrugation profile based on real-time acceleration and forward velocity measurements. Moreover, by using a Gradient-weighted Class Activation Mapping (Grad-CAM) technique, it is shown that the CNN-1D can distinguish various regions, including the transition from damaged to undamaged regions and one-sided or two-sided corrugated regions, while predicting corrugation. In summary, the results of this study reveal the potential of data-driven techniques such as CNN-1D in predicting rails’ corrugation using online data from the dynamics of the rolling-stock, which can lead to more reliable and efficient maintenance and repair of railways.

## 1. Introduction

Corrugation is a common defect in rail contact areas with important implications in safety, serviceability, and cost [1]. Even when the depth of the waves is typically less than 1 mm, rail corrugation can severely impact passenger comfort by causing abnormal accelerations and loud noises. In addition, it is one of the primary reasons for damage and wears both the wheels and rails, leading to higher maintenance costs and service disruption [2,3]. According to numerous researchers, the root cause of this defect lies in the self-excited vibrations and resonance of the wheel–rail system. However, this self-excited vibrations perspective is not universal, and many parameters such as initial regularity, bending mode, torsional vibration, and material mechanics (rail elastic modulus, rail Poisson’s ratio, stiffness of fasteners, etc.) contribute to corrugation generation [2,4,5,6].

Since 1895, many attempts [7,8,9] have been carried out on laboratory experiments, field tests, and analytical and numerical models to predict corrugation [5]. Before 1970, the research on this topic focused on statistical and experimental analysis as well as aspects related to rail material properties, friction, temperature variations, and other relevant factors [10]. After 1970, researchers started to use analytical and numerical models to understand the impacts of resonance, plastic deformation, and residual stress, among other factors, on railway corrugation [10,11,12].

For example, Igeland and Ilias (1997) [13] presented a non-linear model to predict rail corrugation on short-wavelength longitudinal irregularities. They applied various non-linear factors, such as the shift of contact point and the distribution of wear within the contact area. In 2003, Nielsen [14] introduced a numerical model for analyzing rail-head corrugation on straight rails based on dynamic train–track interaction. He concluded that longitudinal slip in the wheel–rail contact interface played a key factor in causing this damage, and his results showed a strong agreement when compared to field observations. Sun and Simpson (2008) [15] carried out numerical modeling and laboratory experiments, which revealed that a tighter curve can cause increased creepage between the wheel and rail, thus causing the wheel stick-slip process, which is a significant factor in the development of corrugation.

Various studies have attempted to predict corrugation growth through frequency domain analysis, mainly using Fast Fourier Transform (FFT) models concentrating on the interaction between wheel and rail at various speeds [16,17,18]. Employing a field investigation and Finite Element (FE) strategies, Chen et al. (2020) [19] determined the wavelengths and frequencies of rail corrugations in three distinct field systems. Their findings indicate that corrugation is nearly certain to occur in the inner rail of a tightly curved track. However, in the case of smooth curved and tangential tracks, they found that friction-induced self-excited vibration does not occur because the creep force is not saturated. In line with this study, Wang et al. (2020) [4] developed an FE model to simulate rail corrugation in Cologne egg fastener sections in small radius curves. The model consists of three sub-models: vehicle, track and wheel–rail contact models. Their wheel–rail contact model comprises the dynamic model of the wheel-track and the non-dynamic model of friction and wear of rail material. After integrating the friction and wear model into the dynamic model, it was revealed that side wear mainly occurs on the outer rail. In contrast, corrugation tends to appear on the inner rail. Despite the success of these physics-based approaches in estimating corrugation, they are time-consuming, computationally expensive, and sensitive to simplifying assumptions and boundary conditions [20]. It is essential to investigate alternative modeling strategies further to address these drawbacks.

Machine Learning (ML) models are popular nowadays due to their excellent capability to tackle complex problems by efficiently handling non-linearities in uncertain systems [21]. In early investigations, ML models, particularly those utlizing images, such as Two Dimensional Convolutional Neural Networks (CNN-2D), have been used to identify corrugation defects and fault segmentation in railways [22,23,24,25]. Nevertheless, there has been a clear need to measure the corrugation geometry properties using tabular data obtained through forward velocity and acceleration sensors, which are becoming increasingly accessible nowadays.

Despite their importance for research, only a few studies have attempted to predict corrugation geometry properties such as depth and wavelength using acceleration and sensor data. Xie et al. (2021). [20] successfully predicted corrugation on metro lines using a novel Deep Learning (DL) model that effectively integrated a One-Dimensional Convolutional Neural Network (CNN-1D) classifier with a hybrid Kriging Surrogate Model (KSM) and Particle Swarm Optimization (PSO) algorithm. In the study, CNN-1D was applied to detect the rail corrugation’s state and classify its wavelength, followed by adopting the KSM-PSO to estimate its depth characteristics. According to their findings, the generated results were in agreement with the observed data. In line with this study, Xie et al. (2023) [26] detected the rail corrugation roughness quantitatively using a modified version of CNN-1D named Recurrent Convolutional Net (RCNet). They used axle box acceleration signals as the input parameters. The model was trained using field data and compared against other DL techniques, such as Long-Short Term Memory (LSTM) and Recurrent Neural Networks (RNNs). Their findings demonstrated that RCNet was superior to other models in accurately predicting corrugation roughness.

The insights gained through ML methods have proven to be valuable in predicting corrugation on railways. However, an important research challenge remains: predicting railway corrugation properties using kinetic data from rolling stock collected during normal railway operation. In this context, this paper aims to address the following objectives:develop a deep learning model to accurately reproduce the rail’s corrugation profile at different forward velocities using kinetic data from the vehicle;investigate the critical railway regions that are important for corrugation prediction by DL models.

In particular, the proposed modeling strategy employs forward velocity and acceleration data in the x (forward) and z (vertical) directions from the vehicle, which are commonly available in real railway systems.

The corrugation was predicted using CNN-1D, and the Gradient-weighted Class Activation Mapping (Grad-CAM) analysis was then used to gain a better understanding of the model’s performance. The CNN-1D model is trained using acquired data about the vehicle’s acceleration and velocity collected at a laboratory scale in the railway research group of the University of Seville. The results obtained confirm the suitability of the proposed data-driven modeling approach in predicting the complete rail’s corrugation profile using just kinetic data from the vehicle.

The remainder of this paper is structured as follows. Section 2 presents the research methodology, which includes a detailed description of the experimental setup and preprocessing techniques, the DL technique used, the design of the model architecture, and hyper-parameter selection. Section 3 will present the main findings, discuss their implications, summarize limitations, and suggest future research directions. Finally, conclusions are provided in Section 4. The research methodology of this study is illustrated in Figure 1.

## 2. Material and Methods

### 2.1. Description of Experimental Setup

This section explains the experimental setup used to gather the datasets. Considering the difficulty of carrying out experiments on actual railroad vehicles and tracks, the railway research group of the University of Seville built a 1:10 scaled track facility, comprising a 90 m long scaled track and an instrumented scaled vehicle [27]. The 1:10 scaled track facility where the experiments were conducted is presented below. The experiments were carried out on a 5-inch wide and 90 m long scaled track built on the roof of the School of Engineering of the University of Seville. Figure 2 shows the plan view of the scaled track: an aerial photograph (Figure 2a) and a schematic plan view of the track center line (Figure 2b).

The designed track includes straight regions, transitions, and curves. The red rectangle highlights the straight corrugated region in Figure 2b. This region has a length of 3.6 meters (m), starting at a track length *s* = 54.9 m and finishing at *s* = 58.5 m. Both rails in this region have been machined to include a realistic corrugation profile. The corrugated profile given to the surface of the rail is based on a combination of four harmonic waves of 5, 10, 20, and 30 mm wavelength with amplitudes of 30, 45, 60, and 75 microns, respectively (see Figure 3). As there is no phase difference between the four waves combination, it results in the 60 mm periodic irregularity shown in Figure 3. This profile (60 mm long) has been repeated consecutively until reaching the total length of 3.6 m.

Figure 4a shows a detail of the corrugated region machined in the scale track. In addition to corrugation, the track has long wavelength irregularities affecting vehicle dynamics. However, since this study aims to predict rails’ corrugation, the effects of the long wavelength irregularities on the accelerometer signals are subtracted by filtering the signals with a high pass filter.

The scaled railway vehicle used in this work was designed and built by the railway research group of the University of Seville. As shown in Figure 4b, the scaled vehicle is a single bogie consisting of two rigid wheelsets, a primary suspension with eight helical springs connecting both wheelsets with the bogie frame. To register the acceleration in the axle box, two uni-axial piezoelectric accelerometers have been installed in the left axle box in the front wheelset of the vehicle in longitudinal (x) and vertical (z) directions. Additionally, one high-precision encoder registers the rotation of the wheels, from which the distance travelled by the vehicle (*s*) and its forward velocity (*V*) is calculated. All the experimental data are acquired and synchronized by a data acquisition system mounted on the vehicle at an acquisition rate of 5000 Hz. The following paragraphs provide a brief explanation of the monitoring framework designed to track degradation in this study.

The test bench to monitor track degradation involves an automated inspection vehicle equipped with various advanced sensors and measurement tools. The vehicle comprises two main parts: the body and the measuring axis, which are connected by a spherical joint to allow for rotation isolation. The body has a traction system with rigid wheels powered by a brushless electric motor and direct gear transmission, ensuring the axis stays centered on the track. The measuring axis, the main component of the vehicle, includes a linear guide and two moving carriages. It is equipped with a precision encoder, inductive sensor, Inertial Measurement Unit (IMU), inclinometer, and Linear Variable Differential Transformer (LVDT) to measure track gauge variation accurately. A NImyRIO-1900 controller (National Instruments, Austin, TX, USA) and a mini-PC with 3G connectivity are used to control the vehicle and acquire sensor data. The cloud-based data handling and processing framework is built on Microsoft Azure, following the RAMI 4.0 hierarchical model. At Level 0, physical assets and processes are managed by sensors and devices connected via PLCs, Fieldbus, distributed systems, or servers, with data communicated to the central system using Azure IoT Edge for real-time data transmission. At Level 1, data ingestion and processing occur through Azure Data Lake Storage for long-term data retention and Azure Data Explorer for short-term analytics. Levels 2 and 3 involve model and digital twin management using Azure Machine Learning and Cosmos DB to develop predictive models and Digital Twin Definition Language (DTDL) to define digital entities. Finally, Level 4 facilitates user interaction through Power BI, providing real-time decision-making support. Figure 5 illustrates the test bench configuration used in this study to monitor track degradation.

### 2.2. Description and Preprocessing of the Gathered Datasets

To predict railway corrugation, laboratory experiments mentioned in the previous section measure three input variables: forward velocity (*V*), acceleration in the z-direction ($A_{z}$), and acceleration in the x-direction ($A_{x}$) [20]. To ensure no variable dominates due to its scale, $A_{z}$ and $A_{x}$ are normalized by $V^{2}$. It is then assumed that the corrugation depends on a group of dynamic factors as follows:(1)$$Corrugation=f\left(V,\frac{A_{x}}{V^{2}},\frac{A_{z}}{V^{2}}\right)$$

The input and output variables are preprocessed using the moving Root Mean Square (RMS) to improve the signal quality and mitigate noise. Utilizing the RMS method to preprocess the data is useful in real-world scenarios where raw signals, such as acceleration and velocity data, are affected by interference from vibrations of other mechanical components and data sequences are not perfectly aligned in time. The RMS method aids in reducing noise and temporal misalignment by averaging the squared values of the signal over a specific time window, thereby smoothing out short-term fluctuations and emphasizing the underlying trend in the data. It is mathematically expressed as follows:(2)$$x_{\mathrm{RMS}}\left[i\right]=\sqrt{\frac{1}{N}\sum_{k=i-N+1}^{i}x^{2}\left[k\right]}$$
where $x_{\mathrm{RMS}}\left[i\right]$ refers to the RMS value at the *i*th point, *N* identifies the number of data points, and x[k] indicates the value of the signal at the *k*th point. These input variables were gathered at V=0.50 m/s and V=1.00 m/s. Table 1 shows the input variable range for each velocity.

Figure 6 illustrates a comparison between the original and the RMS of the input and output signals at V=1.00 m/s, which offers a clear vision of the data transformation.

### 2.3. Deep Learning Predictive Models for Corrugation Prediction

Various deep learning architectures, such as Recurrent Neural Networks (RNNs) and Long Short-Term Memory (LSTM), were initially adopted and optimally configured to address the problem. However, results showed poor generalization due to the complexity of the problem and due to the fact that the past values of the target variable (corrugation) cannot be used as input here, since real-time measurements of corrugation are not available in real scenarios. This issue made it impractical to use an autoregressive component approach in time series using RRNs and LSTM, which relies on past values of the dependent variable (target) as input. Instead, one-dimensional convolutional neural networks (CNN-1D) showed superior predictive capability for this problem and were adopted in this research.

Convolutional Neural Network 1D (CNN-1D) is a powerful deep learning model designed to deal with the difficulties of time series regression [28]. This capability is particularly important because time series data often contains complex temporal patterns, noises, and outliers that can challenge traditional machine learning methods to capture the patterns accurately [29]. The CNN-1D model is a modified version of CNN-2D that can adjust convolutional layers typically used in videos and images to 1D sequences [30,31]. This capability allows the CNN model to handle temporal data, making it a suitable choice for predicting one-dimensional sequences at each time step [32]. Further details of the CNN-1D model can be found in [30].

This study adopts the triple-layer CNN-1D architecture, which comprises three consecutive convolutional layers and two dense layers. Each convolutional layer has appropriate kernel sizes and is equipped with Rectified Linear Unit (ReLU) activation functions to enhance the learning process. After the convolutional layers, the outputs of these layers are flattened and then processed through two fully connected layers, which use linear activation functions. Finally, the output of the last layer is the prediction of corrugation. The architecture of the developed CNN-1D is illustrated in Figure 7.

This figure shows the process of passing input data through multiple convolutional layers to extract patterns. The flattening layer then converts the output from the convolutional layers into a one-dimensional vector, preparing it for input into the fully connected layers. In a fully connected layer, each neuron in one layer is connected to every neuron in the next layer. The number of neurons or nodes in each layer is denoted by the term “unit”. This is the final step in the processing to generate the output. Each layer has different hyper-parameters, which are explained in detail in Section 2.3.1.

#### 2.3.1. The Training Process and Hyper-Parameter Selection

This section discusses the selection of hyper-parameters to clarify how we improve the model’s performance and ensure its reliability. The kernel unit is the core of the CNN-1D model used for time series analysis; therefore, the accurate selection of the number of kernels and their sizes plays a crucial role in the model’s efficiency. Capturing the features and patterns relied on the number of kernels. Increasing the number of kernels when the data is insufficient can lead to overfitting issues [33]. Determining the optimal number of kernels is crucial based on task complexity, network depth, and computational resources [34].

Kernel size refers to the window length that convolves over the data. The size of this window can be different for each convolutional layer. The larger kernel size leads to capturing the broader pattern, while the smaller ones are better for particular details [35]. In this study, various numbers of kernels and kernel sizes were identified for the proposed model. The model employed 256, 128, and 64 kernels to predict corrugation in three consecutive CNN layers. The size of the kernels was set to 4 for the model across different forward velocities. The size of the convolution kernel is fixed at 4 to balance the trade-off between capturing sufficient features and computational efficiency. In practical applications, the choice of convolution kernel size depends on the input data’s specific characteristics and the desired detail level. It is often beneficial to experiment with different kernel sizes and even design convolutional layers with varying kernel sizes to capture diverse features at multiple scales. To tackle the dynamic challenges, such as shifting between observed and predicted results, a trial and error strategy has been applied to arrive at the aforementioned hyper-parameters. Our trial-and-error approach was based on the initial ranges of hyper-parameters suggested by tuning tools such as GridSearchCV, and these parameters were refined through iterative experimentation. This method enabled us to make a balance between maximizing model accuracy, as measured by performance metrics, and improving its ability to generalize across the complex and diverse railway conditions within the data, while also considering computational efficiency.

Stride and padding are other hyper-parameters in the proposed model that play essential roles in the quality of our results. Stride identifies the step size of the kernel over data, while padding (adding zeros at input data edges) guarantees that the kernel applies across all data points. This study determined stride as 1 for the proposed model at different forward velocities, and padding was set as valid. Valid padding limits the slides over the original input data without going beyond bounds. One of its benefits is reducing the convolved features and processing costs.

The Adam optimizer has been used to minimize the Mean Square Error (MSE) loss function to determine the model’s trainable parameters [36]. This study used the “ReduceLROnPlateau” learning rate to fine-tune models by tracking the changes in validation loss [37]. The learning rate changes automatically if validation loss does not reduce for several pre-defined epochs (patience). This dynamic tuning prevents models from getting trapped in local minima.

In order to train the data, the dataset for each velocity was divided into three sub-datasets: training, validation, and testing. The proportions for each sub-dataset were determined through a series of trial-and-error experiments that aimed to improve the overall performance. This division enables accurate evaluation of the models’ performances. Table 2 represents the datasets’ characteristics.

#### 2.3.2. Evaluation Metrics for Model Performance

To better evaluate the predictive abilities of a model, it is crucial to use performance metrics to determine whether the model fits well. The Mean Absolute Percentage Error (MAPE) gives insights into the prediction accuracy, calculated as the mean of all absolute percentage errors between the predicted and observed values. A low value of the MAPE metric indicates a high accuracy of prediction. The Root Mean Square Error (RMSE) was also used, quantifying the square root of the average squared residuals between the predicted and observed values in the dataset. This metric is essential for identifying how well a model fits a dataset. A lower RMSE value signifies a better fit, meaning the model’s predictions are closer to the observed values. Lastly, the Mean Absolute Error (MAE) was calculated, which provides an arithmetic average of absolute errors between predicted and observed values. This metric is beneficial in contexts where the magnitude of errors is of primary concern. These metrics give us a general view of model accuracy and reliability, ensuring a robust evaluation framework for the models in question.

To shed light on how the proposed CNN-1D model maps relationships between inputs and outputs when predicting corrugation at different forward velocities, the Gradient-weighted Class Activation Mapping (Grad-CAM) visualization technique is employed in this study [38]. This technique helps users understand which regions within the data contributed more to generating the final results [39].

To perform Grad-CAM analysis, the first step is to select the last convolutional layer of the neural network. From this layer, the output of each kernel (filter) is extracted, producing feature maps that capture activations. These activations represent the filters’ responses after processing the input data, highlighting patterns and features across the input. Grad-CAM measures the gradient between the predicted results and the feature maps of the last convolutional layer. These gradients are then averaged to obtain weights (activation intensity) $a_{i}$ for each feature map $A_{i}$. These weights indicate how much each feature map contributes to the final prediction. The Grad-CAM heatmap is generated based on these weights. Mathematically, it is represented as
(3)$$H_{\mathrm{Grad}\text{-}\mathrm{CAM}}^{i}=\mathrm{ReLU}\left(\sum_{i}a_{i}A_{i}\right)$$
where $H_{\mathrm{Grad}\text{-}\mathrm{CAM}}^{i}$ represents the Grad-CAM heatmap for predicting value $y^{i}$. $A_{i}$ denotes the feature map from the last convolutional layer, which captures spatial activations across the input data and reflects patterns in the input data. The term $a_{i}$ quantifies the activation intensity or importance of each feature map $A_{i}$ for predicting $y^{i}$. Higher activation intensities indicate a greater influence of the corresponding feature map on the model’s decision, providing insights into critical regions of the input data and contributing to predictions. It is calculated using gradients as
(4)$$a_{i}=\frac{1}{Z}\sum_{t}\frac{\partial y^{i}}{\partial A_{i}\left(t\right)}$$
where $\frac{\partial y^{i}}{\partial A_{i}\left(t\right)}$ measures how sensitive the predicted value $y^{i}$ is to changes in each element $A_{i}\left(t\right)$ of the feature map $A_{i}$ at time step *t*, and *Z* is a normalization factor equal to the total number of time steps *t* in the one-dimensional feature map $A_{i}$. Finally, the ReLU (Rectified Linear Unit) function is applied to the sum of weighted feature maps $a_{i}A_{i}$. ReLU focuses on positive contributions, ensuring that only positive values contribute to the Grad-CAM heatmap $H_{\mathrm{Grad}\text{-}\mathrm{CAM}}^{i}$. For further details on this technique, refer to [40].

The DL model and procedures detailed above form the foundation of the current study. The results are presented and examined in the following results section.

## 3. Results and Discussion

### 3.1. Evaluation of CNN-1D Models’ Performance

This section aims to assess the performance of the proposed methodology to better understand its efficiency and limitations in the context of railway corrugation prediction. To avoid overfitting problems, an early-stopping criterion has been applied by utilizing varying epochs (10, 15, 25, 35, 50, 100, and 150) and patience levels (5, 10, 15, 25, and 50) [41]. After trial and error, it was found that 15 epochs are optimal for V=0.50 m/s and 25 epochs are optimal for V=1.00 m/s, with a patience level of 10. To assess the robustness of the model, it was run 5, 10, 15, 20, 25, and 30 times, and after each set of runs, the global average of MAE was calculated. This systematic repetition minimizes variability and provides a clearer understanding of the model’s performance. In this study, the best MAE value of the model was obtained after 15 runs. Beyond this point, further runs did not improve MAE values.

The global average values of different performance metrics, including the average of Mean Absolute Percentage Error of the 15 individual runs ($MAPE_{ave,15}$), the Mean Absolute Error of the 15 individual runs ($MAE_{ave,15}$) and the Root Mean Square Error of the 15 individual runs ($RMSE_{ave,15}$), were employed to achieve meaningful insights into the best model architectures. Table 3 shows the detailed results for the CNN-1D model for V=0.50 m/s and V=1.00 m/s.

It is clear from Table 3 that the CNN-1D model performed satisfactorily in both training and test stages at different forward velocities. Figure 8a shows the model’s results at V=0.50 m/s in the testing stage. This figure presents a side-by-side comparison of observed (red dashed line) and predicted (black line) corrugation values, helping to display their differences and similarities.

In addition, to have better insights into the distribution patterns and any possible discrepancies between the observed and predicted corrugation values, a residual histogram has been illustrated in Figure 8b, which represents the distribution of errors in the results by the CNN-1D. The X-axis represents the range of residual errors, computed as the difference between predicted and observed corrugation values. The Y-axis shows the frequency of each residual error value expressed as percentages.

As evident from Figure 8b, at V=0.50 m/s, most data points cluster around zero in this histogram, which means the generated results have minimal mean errors, indicating high accuracy.

The prediction results for V=1.00 m/s are shown in Figure 9a. As with the results for V=0.50 m/s, Figure 9b indicates a centered distribution of errors around zero, proving that the CNN-1D model’s results are generally unbiased for that velocity.

### 3.2. Grad-CAM Analysis

As mentioned earlier in Section 2.3.2, to shed light on the mapping relationship process of CNN-1D to predict corrugation, the Gradient-weighted Class Activation Mapping (Grad-CAM) technique was used to gain a better understanding of which regions of the track played crucial roles on the model’s prediction. The activation intensity was calculated at the last convolutional layer of CNN-1D for V=0.50 m/s and V=1.00 m/s. Then, the global average of activation intensity was obtained for both forward velocities. Finally, the Grad-CAM heat maps were generated, as shown in Figure 10a,b. Pure purple indicates a value of ‘0’, representing the track area with minimal influence on the CNN-1D prediction. In contrast, red marks the maximum value of ‘1’, indicating the regions in the track with the highest impact on generated results.

To better interpret these Grad-CAM heatmaps effectively, Figure 11c illustrates the visual representation of measured corrugation profiles. Our system comprises two profiles, each with two undamaged regions, two one-sided corrugated regions, and one two-sided corrugated area. It is important to note that the output values were a combination of corrugation values from both profiles.

Based on the Grad-CAM heatmap shown in Figure 10a for V=0.50 m/s, the model did not assign significant weights to data in the undamaged regions. The coordinates for these regions are between [54.5 m, 54.8 m] and [58.6 m, 59 m] and are represented by pure purple. There is a significant fluctuation in the color spectrum at the first transition region, where the vehicle entered the first corrugated region. This transition region is located around 54.8 to 54.91 m. This region is represented by the label A in Figure 11a,c. Another considerable change in the color spectrum appeared around the region in 55.6 to 55.74 m, which is the starting point of a two-sided corrugated region. Finally, a significant color swing was revealed at the last transition point between damaged and undamaged regions in [58.4, 58.5 m]. The labels B and C were given to the last two regions discussed, as shown in Figure 11a,c.

At V=1.00 m/s, the Grad-CAM heatmap (Figure 10b) shows that both undamaged regions play a significant role in the model decision process. A considerable color shift appeared at the entrance to the damaged region, represented by the label D at V=1.00 m/s. Another marked change in the color spectrum was observed at the center point of the railway, represented by the label E.

These findings reveal that the CNN-1D also detects the second transition region between two-sided and one-sided corrugated regions. This specific transition region identified by the label F can be found at a coordinate of approximately 57.8 m. The final significant fluctuation in the color spectrum, represented by the label G, was observed at the last transition region from a one-sided corrugated to an undamaged region, located near the region C.

### 3.3. Limitations and Future Directions

This research has faced several challenges, particularly regarding the data shifting due to laboratory experiment conditions and the process of hyper-parameter tuning. These challenges shed light on the complexities of using deep learning models in real-world scenarios such as railway maintenance. The influence of the dynamic object’s vibrations before and after passing the sensors can cause data shifting, resulting in discrepancies between inputs and outputs. However, the current study tackles this challenge by post-processing the data using the RMS method and carefully choosing the appropriate method and size for sliding windows.

The tuning of hyper-parameters is also a challenging stage. In this study, selecting the appropriate kernel size and learning rate proved crucial in solving data-shifting and reducing generalization errors. The failure of the models to capture the damage patterns at high forward velocities due to fewer data points emphasizes the need for more laboratory experiments.

In future works, physics-based models and data-driven techniques shall be fused to enhance the accuracy and efficiency of predictions for both low and high velocities. This development helps us exploit both approaches’ potential to improve model predictability. Furthermore, the corrugation predictions can be integrated into a higher-level railway maintenance model to analyze the impact of different levels of damage on the overall railway operation and maintenance.

## 4. Conclusions

The modelling of vehicle-track dynamics is still a fundamental challenge in the railway industry. In this sense, incorporating the new advances in artificial intelligence and data analysis opens a new door that complements the results of physics-based models and leads to a better understanding of degradation processes and their causes. This work demonstrates how new artificial intelligence techniques can be useful in solving challenging engineering problems, such as railway corrugation prediction, which has been partially unsolved using physics-based methods. The present study evaluates the capability of CNN-1D to reproduce corrugation damage in railway systems based on vehicle kinetic data taken at two different forward velocities. The study uses the moving Root Mean Square of forward velocity and acceleration in the longitudinal and vertical directions, normalized by the square of velocity, as input variables to predict corrugation. The findings reveal that the CNN-1D model can reproduce the corrugation profiles with relatively high accuracy.

Grad-CAM analysis was also applied to determine regions with a higher impact on predicted results. The Grad-CAM analysis revealed CNN-1D’s capabilities to distinguish between different railway regions, such as undamaged as well as one-sided and two-sided damaged regions. Regardless of informing the model about various transition regions on the railway, the CNN-1D model successfully detected and assigned different activation intensity values to data in transition regions to predict corrugation, which shed light on its robust internal decision-making process to map the relationship between inputs and outputs.

The results of this study contribute to enhancing proactive maintenance in railways by accurately predicting rail corrugation and detecting damage during normal railway operation. Specifically, the proposed predictive corrugation modeling approach enables monitoring of the state of degradation of extensive railway segments and even the entire network since it is based on kinematic data collected during operation. This predictive modeling can also be integrated within a whole-system condition-based maintenance modeling scheme, employing Petri nets or similar methods to support anticipated and data-driven maintenance decision making.

## Figures and Tables

**Figure 1 sensors-24-04627-f001:**
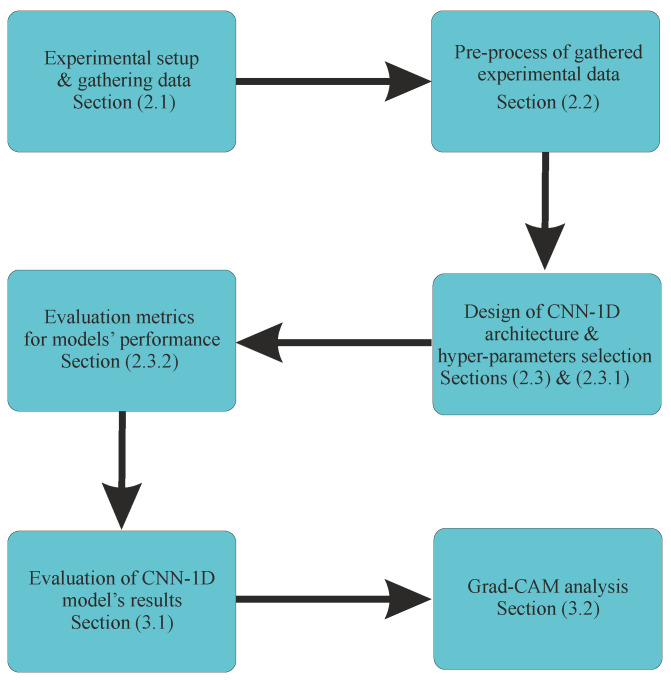
Process of the developed methodology for the prediction of railway corrugation using CNN-1D.

**Figure 2 sensors-24-04627-f002:**
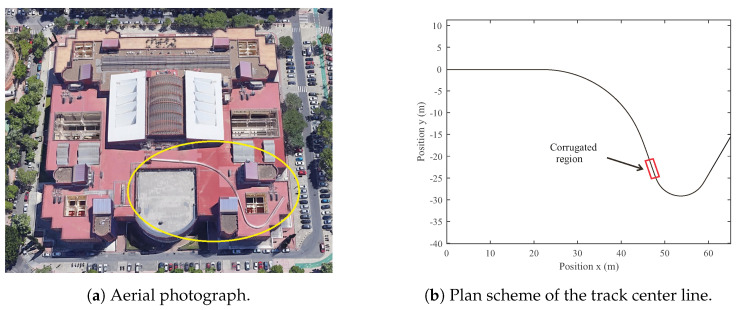
Plan view and geometry details of the scaled track.

**Figure 3 sensors-24-04627-f003:**
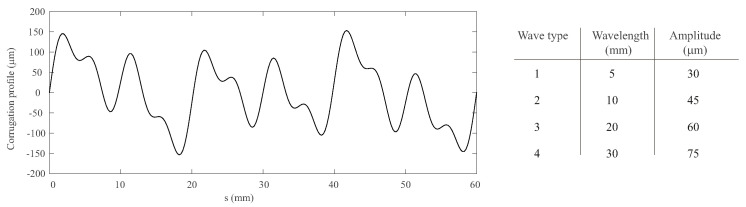
Profile of the designed corrugation.

**Figure 4 sensors-24-04627-f004:**
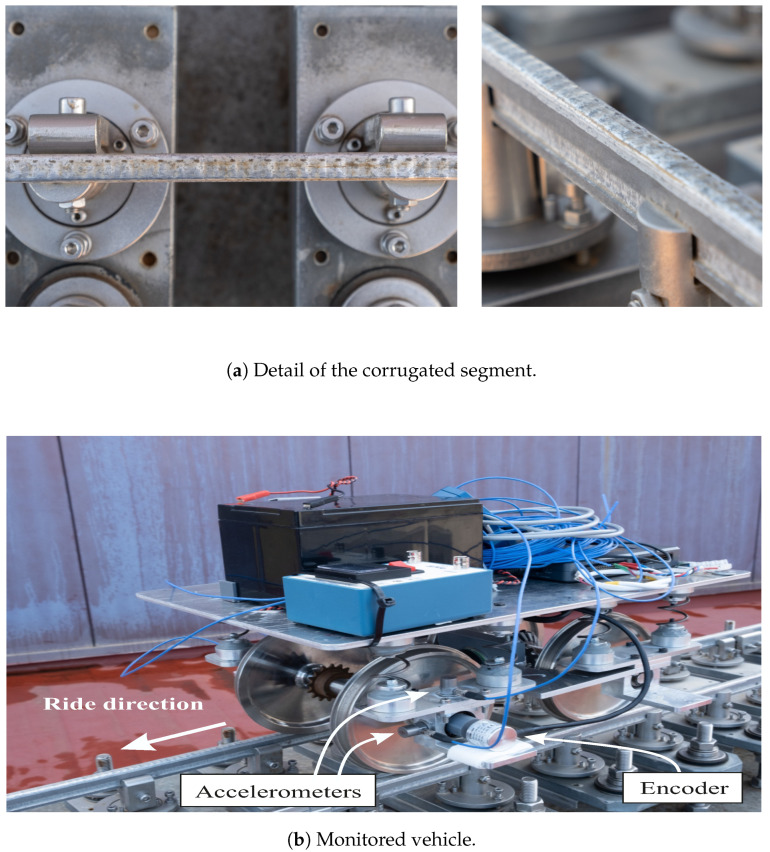
Experimental setup details.

**Figure 5 sensors-24-04627-f005:**
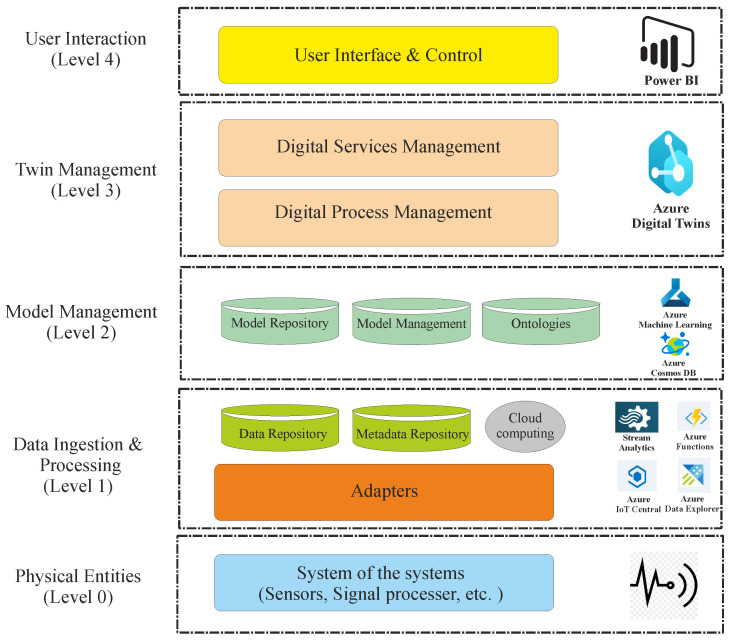
Flowchart of the test bench for monitoring track degradation.

**Figure 6 sensors-24-04627-f006:**
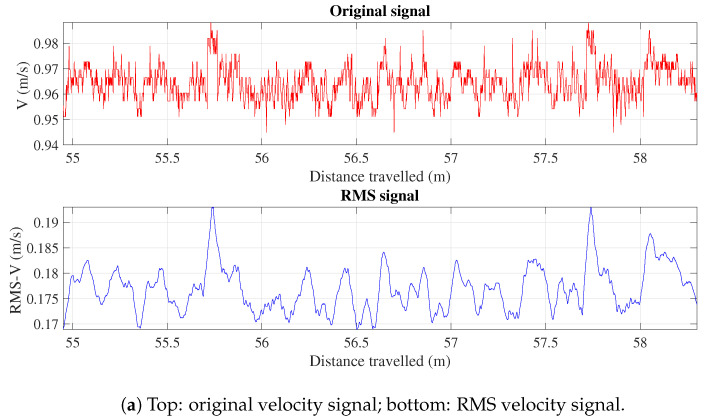
Comparative analysis of original signals and their corresponding RMS transformation for V=1.00 m/s.

**Figure 7 sensors-24-04627-f007:**
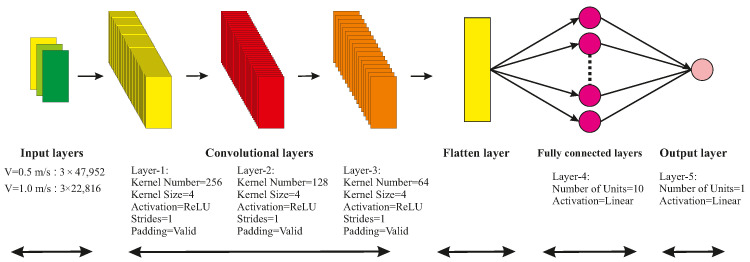
Architecture of the proposed CNN-1D model.

**Figure 8 sensors-24-04627-f008:**
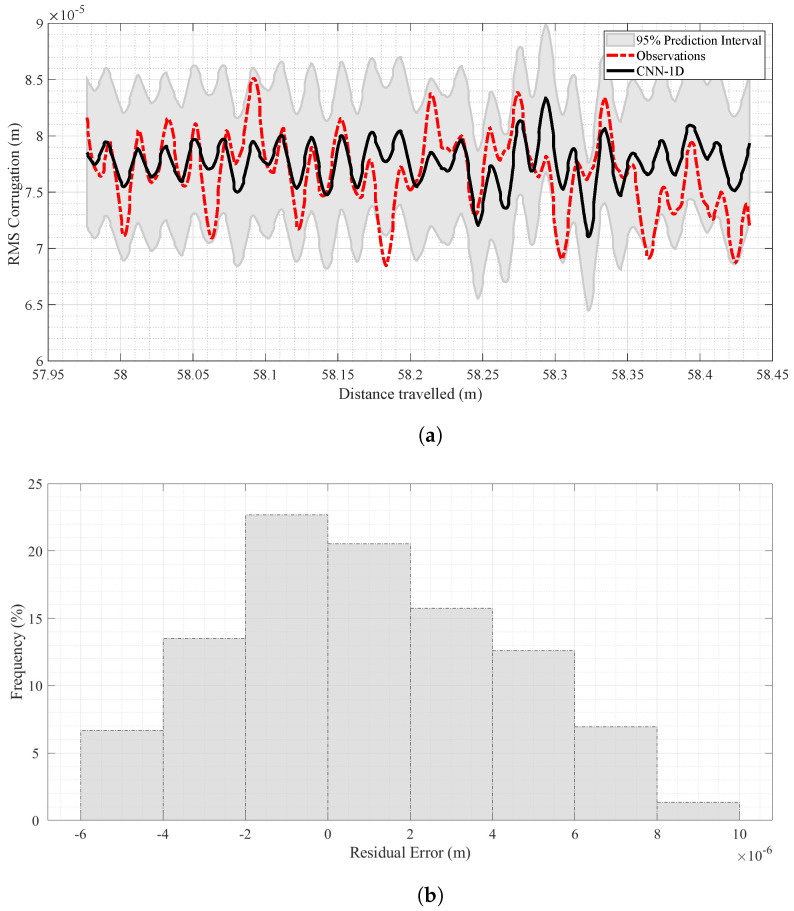
Analysis of predicted and observed data at V=0.50 m/s in the test stage. (**a**) Corrugation prediction results at V=0.50 m/s in the test stage. (**b**) Histogram of residual error for the CNN-1D model in the test stage at V=0.50 m/s.

**Figure 9 sensors-24-04627-f009:**
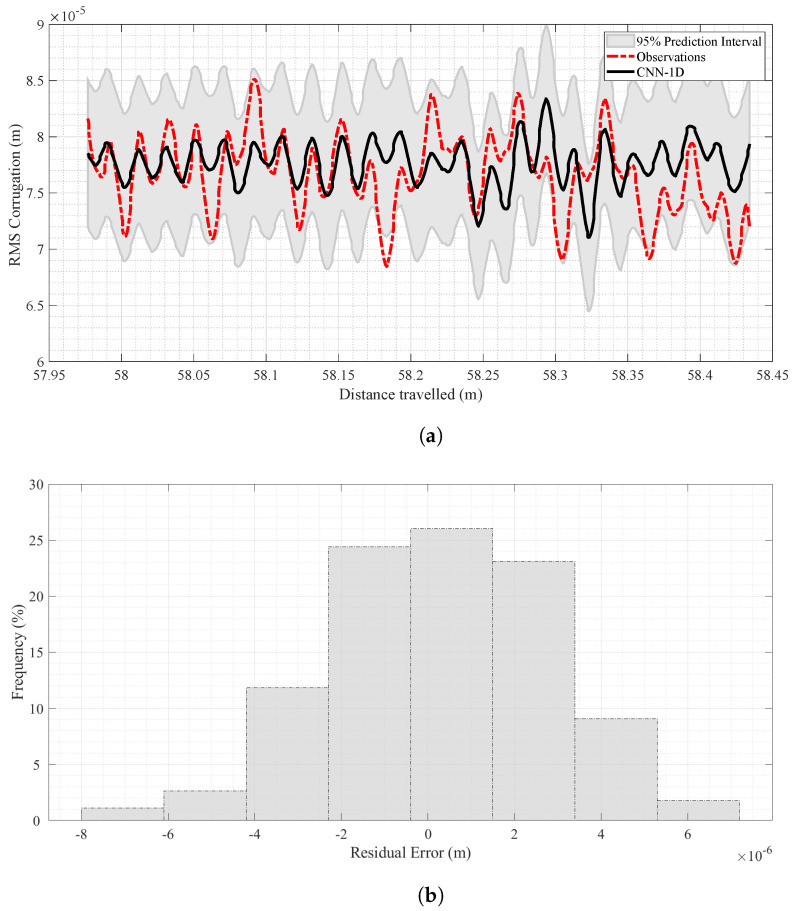
Analysis of predicted and observed data at V=1.00 m/s in the test stage. (**a**) Corrugation prediction results at V=1.00 m/s in the test stage. (**b**) Histogram of residual error for the CNN-1D model in test stage at V=1.00 m/s.

**Figure 10 sensors-24-04627-f010:**
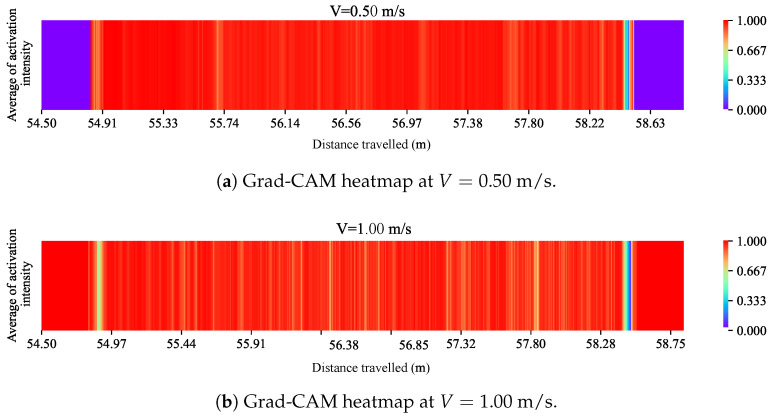
The Grad-CAM heatmaps for V=0.50 m/s and V=1.00 m/s.

**Figure 11 sensors-24-04627-f011:**
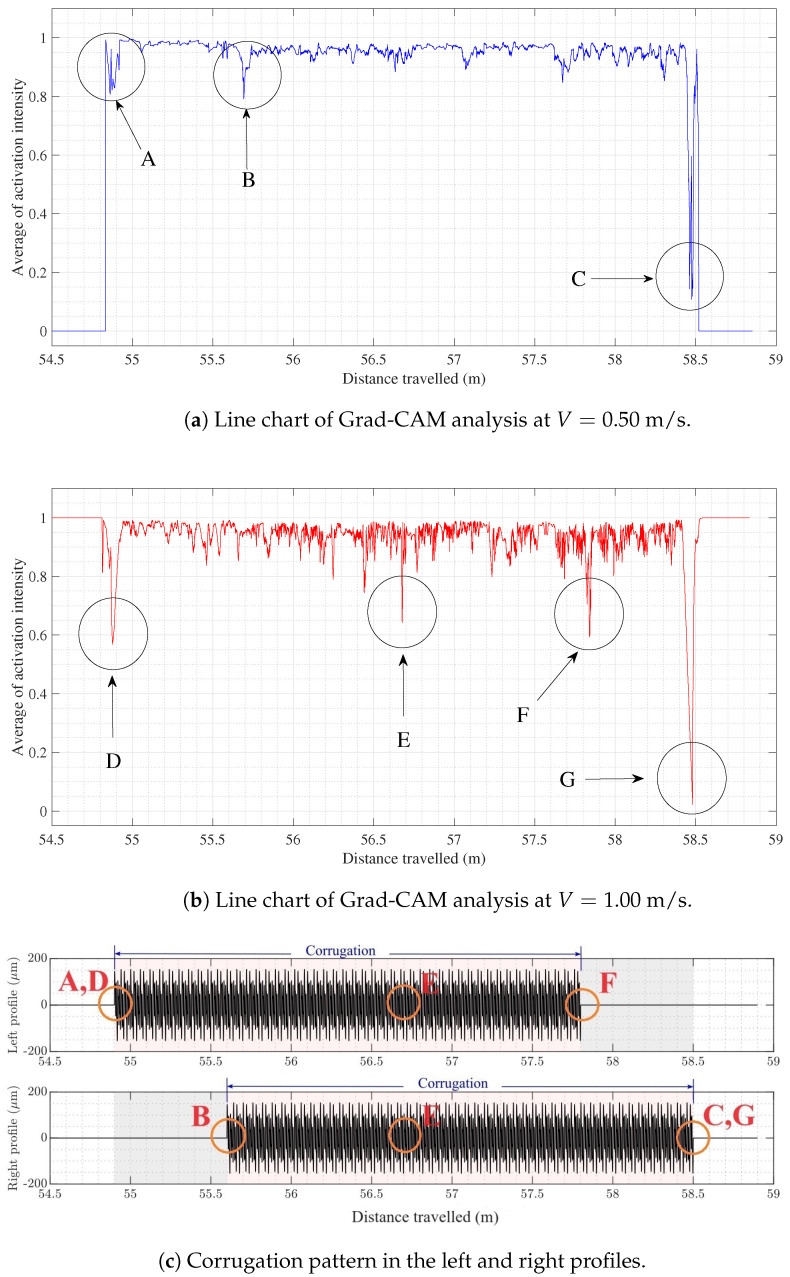
Line charts of Grad-CAM analysis to detect critical regions in CNN-1D regression.

**Table 1 sensors-24-04627-t001:** Range of input variables utilized to predict corrugation.

Original Values						
*V* (m/s)	$V_{\min}$	$V_{\max}$	${\frac{A_{x}}{V^{2}}}_{\min}$	${\frac{A_{x}}{V^{2}}}_{\max}$	${\frac{A_{z}}{V^{2}}}_{\min}$	${\frac{A_{z}}{V^{2}}}_{\max}$
0.50	0.435	0.513	−4.62	7.677	−3.7249	3.297
1.00	0.941	0.988	−3.102	3.527	−2.339	2.1908
RMS Values						
*V* (m/s)	$V_{\min}$	$V_{\max}$	${\frac{A_{x}}{V^{2}}}_{\min}$	${\frac{A_{x}}{V^{2}}}_{\max}$	${\frac{A_{z}}{V^{2}}}_{\min}$	${\frac{A_{z}}{V^{2}}}_{\max}$
0.50	0.07	0.381	0.003569	0.452	0.005149	0.146
1.00	0.168	0.787	0.00649	0.749	0.1031	0.294

**Table 2 sensors-24-04627-t002:** Summary of dataset characteristics.

*V* (m/s)	Number of Data Points	Train %	Validation %	Test %
0.50	47,952	60	20	20
1.00	22,816	60	20	20

**Table 3 sensors-24-04627-t003:** Average of performance metrics for 15 individual runs.

Train Stage			
*V* (m/s)	$RMSE_{ave,15}$ (m)	$MAE_{ave,15}$ (m)	$MAPE_{ave,15}$ (%)
0.50	2.90 × 10^−6^	2.28 × 10^−6^	2.94
1.00	1.52 × 10^−6^	8.93 × 10^−7^	1.14
Test Stage			
*V* (m/s)	$RMSE_{ave,15}$ (m)	$MAE_{ave,15}$ (m)	$MAPE_{ave,15}$ (%)
0.50	3.5 × 10^−6^	2.83 × 10^−6^	3.77
1.00	2.56 × 10^−6^	2.09 × 10^−6^	2.72

## Data Availability

Dataset available on request from the authors.

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
