# Peer review of "Predicting Rail Corrugation Based on Convolutional Neural Networks Using Vehicle’s Acceleration Measurements"

_sensors, 2024, doi:10.3390/s24144627_

Round 1

Reviewer 1 Report

Comments and Suggestions for Authors

In this paper, the research on rail corrugation prediction based on ConNetwork using Vehicle's Acceleration Measurements is carried out, and verified by the scaling model. The overall research has certain theoretical research significance and engineering application value. However, there are still some problems to be corrected.
(1)There are four types of wave abrasion in the test section, the wavelength is 5mm, 10mm, 20mm and 30mm. First, these wavelengths do not correspond to the small radius section on site, how to explain? In addition, the longer the wavelength here is, the deeper the wave depth is. There is a certain difference between this and the site? How to verify the accuracy of the scaling model.
(2)How to combine the research in this paper with the actual research in the field to promote the application.

Comments on the Quality of English Language

(1)It is suggested to optimize the information of pictures in the revised text, such as Figure 4b.

Reviewer 2 Report

Comments and Suggestions for Authors

The paper demonstrate that CNN 1-D has the potential to estimate the corrugation through vehicle vibration accelerations and vehicle speed. The contribution are: 1)developing a CNN 1-D model, and 2) using a model visualization technique(Grad-CAM) to determine the critical railway regions that has important impact on corrugation prediction. The research is interesting.

Some suggestions:

1. On page 6, in practical applications, under what circumstances should we use the RMS method for pre-processing?

2. On page 10, please provide a more specific description of the Grad-CAM method, such as a diagram or formula.

3.In Figure 6, (1) In the two fully connected layers behind the flatten layer, what is the unit, and how fully connected; (2) Why is the size of the convolution kernel fixed at 4? How should we choose the size of the convolution kernel in practical applications, and whether should we design convolution kernels with different sizes.

4. In Figures 10, the blank area in front of figure 10(b) is meaningless. The horizontal axis in figure 10 (b) should be aligned with the horizontal axis in figure 10 (a) .

5. Have the authors considered using recurrent neural networks, such as LSTM and GRU, for corrugation prediction?

Reviewer 3 Report

Comments and Suggestions for Authors

The work is devoted to the use of CNN-1D to solve the problem of determining rail corrugation using a bogie model and measuring acceleration.

In general, the work is some kind of superficial and rather serves as a demonstration of the use of convolutional networks of type  CNN-1D to solve some problems, in this case, how movement occurs on a worn-out rail and what predictions come from this. That is, not as a real case of solving a practical problem, but rather as a case-study for using a neural network and tuning its parameters and smoothing data from inputs in order to obtain more representative outputs.

For the Sensors journal, the reviewer would expect a detailed description of your test bench with its hardware architecture and measurement tools that were used to generate the dataset. At the level of what controllers, sensors and data buses were used.

Also, the work lacks a really detailed analysis of the hyperparameters used for the operation of the neural network and how they are applicable to solving this problem, although methods for their selection have long been proposed, and since the authors are already familiar with the subject area, their choice should also be justified by physical limitations and a set experiments, since the results greatly depend on them.

Minor:

Figure 1 contains highly naive pinctograms.

Figure 2: Aerial photograph (Capital A)

Figure 4: Experimental setup components - not a very good picture caption

Figure 6: poorly explained

Round 2

Reviewer 3 Report

Comments and Suggestions for Authors

The authors made a good cover letter, taking into account comments on the relevance of the work and on the hardware stand and on hyperparameters.

I accept authors' corrections.

The only thing is that for the Sensors journal I would still want a diagram of the hardware architecture of the monitoring vehicle, described in the new text of section 2.1.
